# Genetic Diversity of Historical and Modern Populations of Russian Cattle Breeds Revealed by Microsatellite Analysis

**DOI:** 10.3390/genes11080940

**Published:** 2020-08-14

**Authors:** Alexandra S. Abdelmanova, Veronika R. Kharzinova, Valeria V. Volkova, Arina I. Mishina, Arsen V. Dotsev, Alexander A. Sermyagin, Oxana I. Boronetskaya, Lidia V. Petrikeeva, Roman Yu Chinarov, Gottfried Brem, Natalia A. Zinovieva

**Affiliations:** 1L.K. Ernst Federal Science Center for Animal Husbandry, 60, pos. Dubrovitsy, 142132 Podolsk, Russia; preevetic@mail.ru (A.S.A.); veronika0784@mail.ru (V.R.K.); moonlit_elf@mail.ru (V.V.V.); arinamishina32@yandex.ru (A.I.M.); asnd@mail.ru (A.V.D.); alex_sermyagin85@mail.ru (A.A.S.); liskun@rgau-msha.ru (O.I.B.); ulreeka@gmail.com (L.V.P.); roman_chinarov@mail.ru (R.Y.C.); gottfried.brem@vetmeduni.ac.at (G.B.); 2Timiryazev Russian State Agrarian University—Moscow Agrarian Academy, 49, ul. Timiryazevskaya, 127550 Moscow, Russia; 3Institut für Tierzucht und Genetik, University of Veterinary Medicine (VMU), Veterinärplatz, A-1210 Vienna, Austria

**Keywords:** microsatellites, consensus genotypes, multiple-tube approach, historical DNA, Russian local cattle breeds, genetic diversity

## Abstract

Analysis of ancient and historical DNA has great potential to trace the genetic diversity of local cattle populations during their centuries-long development. Forty-nine specimens representing five cattle breeds (Kholmogor, Yaroslavl, Great Russian, Novgorod, and Holland), dated from the end of the 19th century to the first half of the 20th century, were genotyped for nine polymorphic microsatellite loci. Using a multiple-tube approach, we determined the consensus genotypes of all samples/loci analysed. Amplification errors, including allelic drop-out (ADO) and false alleles (FA), occurred with an average frequency of 2.35% and 0.79%, respectively. A significant effect of allelic length on ADO rate (*r^2^* = 0.620, *p* = 0.05) was shown. We did not observe significant differences in genetic diversity among historical samples and modern representatives of Kholmogor and Yaroslavl breeds. The unbiased expected heterozygosity values were 0.726–0.774 and 0.708–0.739; the allelic richness values were 2.716–2.893 and 2.661–2.758 for the historical and modern samples, respectively. Analyses of *F_ST_* and Jost’s D genetic distances, and the results of STRUCTURE clustering, showed the maintenance of a part of historical components in the modern populations of Kholmogor and Yaroslavl cattle. Our study contributes to the conservation of biodiversity in the local Russian genetic resources of cattle.

## 1. Introduction

Since ancient times, cattle have been an essential element in the subsistence of humankind because they have provided humans with food, clothing, and draught force. Modern animal breeding uses a small number of highly productive commercial breeds originating from a limited number of elite sires [1]. Most of the local breeds cannot compete with commercial breeds owing to their lower productivity and lack of adaptability to industrial production systems. This has led to a drastic decline in the population census size of the majority of the local breeds or to the extinction of several breeds [2]. Most of the local breeds are carriers of unique forms of variability associated with adaptations to geoclimatic conditions, desired milk composition, disease resistance, and other useful breed-related properties [3], which make them an invaluable source of genetic diversity for livestock management in the future [4,5]. A study of local breeds aimed at the identification of ancestral breed-specific genetic components is important for their conservation, sustainable breeding, and development.

The most prominent local breeds of black and white cattle, bred in the Central and Northern parts of European Russia, are Kholmogor and Yaroslavl (Appendix A). They originated from the Northern Great Russian cattle, low productive animals of small size, which are able to survive in harsh environments with poor forage [6]. During the 18th–19th centuries, Kholmogor cattle were well-known in Russia and abroad owing to their high productivity and excellent quality of dairy products. Yaroslavl cattle were highly demanded by peasants and small cattle holders owing to their adaptability, high feed efficiency, and good reproductive ability even in poor forage conditions during winter [7]. The origin of Kholmogor cattle remains questionable: numerous studies have highlighted a high contribution of Holland (Dutch) cattle [3,8], whereas other sources have highlighted only a limited participation of Holland bulls [9,10,11,12,13]. Kholmogor cattle, mainly bulls, were used to improve several other Russian cattle breeds, including Yaroslavl and Novgorod cattle (local cattle population of the Novgorod province of old Russia). The drastic decline in population census size of Kholmogor and Yaroslavl breeds from close to one million individuals each in the 1960s to about 220 and 50 thousand heads in 2015, respectively [14], requires the development of programs for conservation of biodiversity of these Russian local breeds. The efficiency of such programs can be significantly improved by identification and selection of animals which carry the majority of their ancestral genetic components.

Several types of DNA markers have been used for genetic studies of modern cattle populations; the most widely used are mtDNA polymorphisms [15], microsatellites, also known as short tandem repeats (STRs) [8,15], and single nucleotide polymorphisms (SNPs) [12,13,16,17]. Although the analysis of genomes using SNP chips is becoming more widespread [18,19], microsatellite markers continue to serve as a powerful tool for inferring population patterns in non-human systems [20], including cattle [21]. Moreover, microsatellites are the gold standard for parentage testing in most breeding programs of cattle [22].

Analysis of ancient and historical DNA [18,23,24,25], including the specimens stored in museum collections [26,27] and craniological collections [14], has great potential for reconstructing the breed origins and history of local cattle husbandry. The significant methodological obstacle of studying ancient and historical specimens is the retrieval of a sufficient amount of pure authentic DNA that is suitable for molecular genetic analyses [28,29]. The extraction of DNA from the specimens maintained in craniological collections is complicated by the harsh treatment of skulls for deposition, which can lead to a significant degradation of nucleic acids. The recent success in the development of protocols for effective DNA extraction from ancient and historical samples enable the molecular genetic analysis of such samples using different tools [18,30,31,32].

Studies of DNA derived from ancient samples tend to focus on the mitochondrial genome [33], because it is present in hundreds or even thousands of copies in a single animal cell [34]. However, in some cases, a study of mtDNA has a limited ability to infer complex demography of animal species [24,35]. Analysis of microsatellites is a method of choice for the study of such populations of livestock species. Besides, during the long-term application of microsatellites for molecular genetic studies of farm animals, a relatively large number of samples derived from modern representatives of local cattle breeds were genotyped, including those breeds for which whole-genome SNP genotypes were not available or were generated only for a limited number of samples [8,36]. The presence of larger sets of modern genotypic data is important for the study of historical samples because it can help to identify more precisely those alleles that have not been transmitted from historical to modern generations of animals or those with frequencies that are markedly different.

The main problem associated with microsatellite genotyping of DNA derived from ancient and museum specimens are the amplification errors, such as false homozygotes, also known as allelic drop-out (ADO), and false alleles (FA) [37,38], which are slippage artefacts during polymerase chain reaction (PCR) [39]. To obtain reliable genotypes for poor or degraded DNA, a multiple-tube approach, which is based on repeated independent amplifications of each DNA sample, was proposed [37,40]. Comparing to the standard one-tube procedure, the multiple-tube approach has the advantage of providing a quantitative measure of the degree of support for each possible genotype [40].

In the present study, using a multiple-tube approach, we generated microsatellite genotypes for 49 historical samples dated from the end of the 19th century to the first half of the 20th century, representing four native Russian cattle populations of the Central and Northern parts of European Russia, and compared them with modern representatives of the same breeds that have been maintained until now to trace their biodiversity and characterise the relationship between both populations.

## 2. Materials and Methods

### 2.1. Breeds and Samples

Cattle skulls dated from the end of the 19th century to the first half of the 20th century, which are stored in the craniological collection of the Museum of Livestock, named after E.F. Liskun (Moscow Agricultural Academy, named after K.A. Timiryazev), were used as the source of historical samples. A total of 46 historical samples of Russian local cattle populations were included in the study, including Kholmogor (KH_H, *n* = 22), Yaroslavl (YR_H, *n* = 20), Novgorod (NV_H, *n* = 2), and Great Russian cattle (GR_H, *n* = 2). Historical samples of Holland cattle (HL_H, *n* = 3) were also included in the study because of their active importation to old Russia during the 19th century and the first half of the 20th century, and their possible contribution to the formation of Russian cattle breeds. A sample of modern representatives of Kholmogor (KH_M, *n* = 177) and Yaroslavl (YR_M, *n* = 61) breeds was used for comparison. A sample of modern Holstein breed (HS_M, *n* = 152) was used as an out-group.

### 2.2. DNA Extraction

All experiments using the historical samples were performed in the facility of the L.K. Ernst Federal Science Centre, dedicated to studies involving ancient DNA [30]. We used teeth to produce bone powder for DNA extraction of the historical samples. The teeth were recovered from skulls, treated with hydrogen peroxide, and irradiated using ultraviolet light (254 nm). Then, the teeth were sawn lengthwise and bone powder was obtained using a drill. Genomic DNA was extracted from the dentin powder using commercially available kits: Prep Filer™ BTA Forensic DNA Extraction Kit (Thermo Fisher Scientific Inc., Waltham, MA, USA), QIAamp DNA Investigator Kit (Qiagen, Valencia, CA, USA), or COrDIS Extract decalcine (GORDIZ LLC, Moscow, Russia), according to the instructions of the manufacturers with modification on the amount of bone powder (100 ± 5 mg) and lysis conditions (56 °C, 1200 rpm, overnight). To check the sample processing reagents for the possible occurrence of DNA contamination, all procedures of DNA extraction were performed with a negative control tube (“reagent blank”) containing only DNA extraction reagents without sample.

Genomic DNA of modern animals was extracted from tissue or sperm samples using Nexttec columns (Nexttec Biotechnology GmbH, Leverkusen, Germany), according to the manufacturer’s instructions. Animal tissue samples were collected by trained personnel under strict veterinary rules in accordance with the rules for conducting laboratory research (tests) in the implementation of the veterinary control (supervision) approved by the Council Decision Eurasian Economic Commission No. 80 (10 November 2017). The sperm samples were provided by the artificial insemination stations.

The quantitative and qualitative characteristics of the obtained DNA were evaluated by measuring the concentration of double-stranded DNA (dsDNA) using a Qubit™ fluorimeter (Invitrogen, Life Technologies, Waltham, MA, USA) and determining the ratio of the absorption at 260 and 280 nm (OD260/280) on a NanoDrop 8000 instrument (Thermo Fisher Scientific Inc., Waltham, MA, USA).

The risk of genotyping errors decreases if more template DNA of good quality are available for PCR; thus, we set the threshold for concentration of dsDNA at 1 ng/µL and the limits for OD260/OD280 ratio at 1.6–2.0. The samples with DNA concentration lower than the threshold and out of the OD ratio limits were discarded, and new DNA extractions were carried out using an increased amount of bone powder or by replacing the DNA extraction kit.

### 2.3. Microsatellite Genotyping and Detection of Consensus Genotypes

Nine cattle microsatellite loci were selected from the list of microsatellite markers recommended by the International Society of Animal Genetics (ISAG) [41]; the selected loci were: BM2113, BM1824 [42], ETH10 [43], ETH225 [44], INRA023 [45], SPS115 [46], TGLA122, TGLA126, and TGLA227 [47]. These loci are high-polymorphic, carrying 12, 8, 9, 11, 16, 11, 18, 11, and 16 alleles, respectively [22]. The multiplex (9-plex) PCRs were conducted in a final volume of 10 mL in a PCR buffer with 200 mM dNTPs, 1.0 mM MgCl_2_, a 0.5 mM primer mix, 1 unit of Taq polymerase (Dialat Ltd., Moscow, Russia), and 1 µL of genomic DNA. After the initial denaturation (95 °C, 4 min), 35 cycles were performed at the following temperature and time regimens: 95 °C, 20 s; 63 °C, 30 s; and 72 °C, 1 min. The final extension was performed at 72 °C for 10 min. To check the PCR reagents for DNA contaminations, we included negative control (PCR reaction without DNA template) in each PCR experiment. Fragment analysis was performed on an ABI3130xl genetic analyser (Applied Biosystems, Beverly, MA, USA) using GeneScan™-350 ET ROX as a fragment standard. The Gene Mapper software v. 4 (Applied Biosystems, Beverly, MA, USA) was used to determine the fragment lengths. Allele sizes were standardised according to ISAG International Bovine *(Bos Taurus)* STR typing comparison test 2018–2019.

To determine the consensus genotypes, we used a modified multiple-tube approach, proposed by Mondol et al. [48] and used by Modi et al. [49]. After initial multiplex amplification of microsatellite loci in duplicates, we selected only the samples in which at least six loci were successfully amplified (positive multiplex PCRs) for further analysis. For such samples, we performed three additional independent PCR replicates using the same DNA extractions. Thus, each DNA sample was analysed in five replicates. For samples in which less than six loci were amplified or less than four positive PCRs at any locus were received, new DNA extractions and PCR amplifications were carried out as described above.

Based on genotyping results of the five PCR replicates, we calculated the “quality indices” (QIs) for each sample/locus as described by Miquel et al. [50]. The most frequent genotype at each locus was coded as “1”. The presence of genotypes, which were different from the most frequent genotype owing to allelic drop-out (ADO) or false allele (FA), was considered as genotyping errors and was coded as “0”. The threshold value for QI was set at 0.75. For samples with QI value of less than 0.75 at any locus, three additional multiplex PCRs were carried out using the same DNA extraction. Only the samples with a QI of 0.75 and higher at each locus were considered reliable and were selected for further analysis. For samples with QIs < 0.75, new DNA extractions and PCR amplifications were carried out using the above-mentioned multiple-tube approach or were discarded and not used for further analysis. A schematic representation of the genotyping protocol is presented in the Appendix A.

### 2.4. Statistical Data Analysis

Amplification failure for each locus was calculated as the number of positive PCRs divided by the number of replicates. For ADO and FA rate calculations, we used the protocol proposed by Broquet and Petit [51]. The ADO rate for each locus was calculated for heterozygous genotypes (according to corresponding consensus genotypes) as the number of replicates where one allele was lost, divided by the total number of positive PCRs. The FA rates were calculated for homozygous or heterozygous genotypes as the number of PCRs with false alleles divided by the total number of positive PCRs. Overall error rates were calculated as the number of detected genotypes altered from the consensus genotype divided by the total number of positive PCRs. The probability values of correct genotyping (*p*) for each locus were calculated according He et al. [52].

We checked the presence of null alleles using FreeNA [53] and MICRO-CHECKER 2.2.3 software [54]. With FreeNA, the null alleles with frequency < 0.35 were detected in the GR_H group, owing to the small number of samples. The null alleles with frequency < 0.21 were detected in HL_H and YR_M. The use of MICRO-CHECKER showed the presence of null alleles with frequencies < 0.23 in YR_M. We did not observe significant differences in the results using raw and adjusted data corrected by Oosterhout [54] and Chakraborty [55] algorithms realised in the MICRO-CHECKER. Based on these results and the consideration that the frequencies of the null alleles were < 0.50, we decided not to exclude these alleles from subsequent analysis [56].

Observed (*H_O_*) heterozygosity, unbiased expected heterozygosity (*_U_H_E_*), unbiased inbreeding coefficient (*_U_F_IS_*), and rarefied number of alleles (*A_R_*), which measures the contribution of alleles weighted by the sample size [57], were calculated using the R package diveRsity [58]. GenAIEx 6.5 [59] was used to calculate the number and frequency of alleles. Principal component analysis (PCA) was performed using the R package adegenet [60] and visualised in the R package ggplot2 [61]. The data files were prepared in the software environment R3.5.0 [62].

The degree of genetic differentiation of the studied breeds was evaluated based on pairwise Jost’s D values [63] and paired *F_ST_* values [64]. The *F_ST_* and Jost’s D matrices of the pairwise values were used to construct a phylogenetic network using the Neighbour-Net algorithm in SplitsTree 4.14.5 [65].

The genetic structure was investigated using an admixture model with the option of correlated allele frequencies in the STRUCTURE 2.3.4 program [66]. We set a burn-in period to 10,000 iterations followed by 100,000 Markov chain Monte Carlo (MCMC) repetitions for each run. This setting produced consistent estimations that were not significantly altered by a longer burn-in and MCMC. We tested the range of possible clusters (*K*) from 2 to 8 (corresponds to the number of studied populations), performing 10 independent runs for each *K*. STRUCTURE HARVESTER [67] with the Evanno method [68] was used to estimate the most likely number of *K* that explained the sample structure. The program CLUMPAK [69] available at http://clumpak.tau.ac.il, was used to analyse multiple independent runs at a single K value. Average pairwise similarity scores as a measure of the constancy over runs, implemented in CLUMPAK [69], were used to define the population structure.

## 3. Results

### 3.1. Estimation of Consensus Genotypes

The concentrations of dsDNA in historical samples, which were selected for genotyping, varied from 1.04 to 63.40 ng/µL, and OD260/280 ratio ranged from 1.66 to 2.00. Using the multiple-tube approach, we were able to estimate the consensus genotypes for all forty-nine samples analysed for all nine microsatellite loci (Appendix A). The quality of genotyping based on QI values differed between microsatellite loci, varying from QI = 0.965 in TGLA126 to QI = 0.991 in ETH225. The probability values of correct genotyping (*p*) at each locus were less than threshold (*p* < 0.001) (Appendix A). Amplification failure among loci varied from 0.00% in BM2113 to 5.88% in INRA023 with an average of 2.33%. Allelic drop-out was observed in 2.35% of the estimated heterozygous genotypes and varied from 0.90% in TGLA227 to 4.07% in the SPS115 locus. False alleles were identified in 0.79% of the positive PCRs and varied from 0.37% in BM2113 and ETH225 loci to 1.17% in SPS115 and INRA23 loci. Average error rate was 2.59% from the number of positive PCRs with variation ranging from 1.12% in the ETH225 locus to 3.89% in the SPS115 locus (Table 1).

### 3.2. Genetic Variability

In total, ninety-five microsatellite alleles were identified in eight cattle populations, including eighty-one alleles from historical populations and eighty alleles from modern populations. All loci were polymorphic except for BM1824 in historical Novgorod cattle, which could be explained by the small sample size. While loci TGLA122 and TGLA227 were characterised by the highest variability (sixteen and twelve alleles, respectively), loci BM1824, SPS115, and TGLA126 were less polymorphic (five, six, and seven alleles, respectively) (Appendix A).

Details of variability measurement across 439 individuals of historical and modern cattle populations are summarised in Table 2.

We did not observe any significant differences in genetic diversity between the historical and modern populations of Kholmogor and Yaroslavl cattle. The unbiased expected heterozygosity (*_U_H_E_*) values were 0.726–0.774 and 0.708–0.739; the allelic richness (*A_R_*) values were 2.716–2.893 and 2.661–2.758 for the historical and modern samples, respectively. The highest values of *_U_H_E_* and *A_R_* were detected in Great Russian cattle (0.852 and 3.111, respectively), whereas the lowest were detected in Holland cattle (0.644 and 2.533, respectively). We observed a significant deviation from the Hardy–Weinberg equilibrium in heterozygote number, only for historical samples of the Kholmogor breed (heterozygote excess), which has been indicated by a negative value of *_U_F_IS_* (−0.80).

Comparison of historical and modern samples showed that 15 alleles, distributed in the historical population, were lost in the modern population (Appendix A). A total of twelve novel alleles occurred in the modern Kholmogor and Yaroslavl breeds. TGLA122 locus was the most altered, where six and two unique alleles were identified in historical and modern groups, respectively. The most affected in terms of the number of lost alleles were the historical samples of the Yaroslavl breed, where eight such alleles were found.

### 3.3. Principal Component Analysis

PCA of historical and modern cattle populations based on nine microsatellite markers are shown in Figure 1. The first component, which was responsible for 4.298% of the genetic variability, differentiated the Holsteins from all the other studied populations. The second component, which explained 3.368% of the genetic differences, differentiated the modern Yaroslavl and Kholmogor breeds. The historical samples of these breeds are more heterogenic compared to the modern populations and form overlapped clusters with their modern representatives. The historical and modern samples of the Kholmogor breed had the closest position on the Y-axis with Holsteins, which revealed their later differentiation from the Holsteins compared to the Yaroslavl breed.

### 3.4. Genetic Relationship between Populations

Genetic relationships identified between the studied breeds based on both pairwise Jost’s D and *F_ST_* values in most cases were similar (Table 3). Negative values of both Jost’s D and *F_ST_*, which were observed between historical Great Russian and Novgorod cattle, revealed the absence of genetic differences between these populations. Holsteins were the most divergent among the historical and modern studied Russian local cattle populations.

To visualise the genetic distances between the studied populations, we constructed phylogenetic dendrograms, based on both *F_ST_* and Jost’s D pairwise genetic distances using the Neighbour-Net algorithm (Figure 2A,B).

As shown in Figure 2, historical Great Russian and Novgorod cattle formed the join branch on *F_ST_*-based network (Figure 2A) and localised close to each other on the edge of the Jost’s D graph (Figure 2B), which indicated their genetic similarity. The formation of a separate branch by historical Holland cattle can be interpreted as the small participation of Holland cattle in the development of the studied Russian cattle breeds. The neighbour localisation of branches of historical Holland and Kholmogor cattle on Jost’s D network (Figure 2B) suggests a higher contribution of Holland cattle in the improvement of the historical Kholmogor population compared to that of historical Yaroslavl cattle. The positioning of modern Yaroslavl cattle at the end of the branch, formed by historical samples of this breed (Figure 2B), might indicate that the breed improvement during the last century occurred mainly by selection of purebred animals with low contribution from other breeds.

### 3.5. Genetic Structure Analysis

The highest *ΔK* value was observed for *K* = 2 (Appendix A). The average pairwise similarity scores among 10 replicates were 0.998 (*K* = 2), 0.995 (*K* = 3), 0.984 (*K* = 4), 0.987 (*K* = 5), 0.987 (*K* = 6), 0.984 (*K* = 7), and 0.970 (*K* = 8). Results of the STRUCTURE analysis for the number of clusters from 2 to 4 are presented in Figure 3. At *K* = 2, Holsteins are clearly differentiated from historical and modern Russian local cattle populations, thereby indicating their development from different ancestral populations. We observed the visible presence of Holstein-specific genetic components in part of the historical samples of the Kholmogor and Yaroslavl breeds and in Holland cattle. At *K* = 3, the modern Yaroslavl breed was clearly separated from the Kholmogor breed; several modern Kholmogor cattle revealed admixture signals. At this *K* value, the cluster structure of modern Kholmogor and Yaroslavl cattle was similar to that of historical populations of the respective breeds. Analysis at *K* = 4 showed that both modern Kholmogor and Yaroslavl breeds maintained the historical components in their cluster structures.

## 4. Discussion

Owing to centuries of breeding history, local breeds of cattle have formed a unique gene pool that makes them a valuable source of biodiversity [2,3,4,5]. Studies of ancient and historical DNA from domestic animals can shed new light on the process of domestication [18] as well as on the history of local animal husbandry [23,24,25]. So far, most studies of historical and ancient DNA of livestock species including cattle targeted mitochondrial DNA (mtDNA) [70,71,72,73,74], which has limited power in characterising the relationships between modern and ancient livestock populations, especially when the history of the populations has been affected by human-driven migrations, admixture, and intensive sex-specific breeding practices [75]. During the 18th–19th century in Russia, the most widely used breeding praxis in cattle husbandry was the exchange of bulls between landlords and bull import from abroad [76]. During the last few decades, crossbreeding using elite sires has been performed to improve Russian local cattle, including the Kholmogor and Yaroslavl breeds. Thus, we used nine microsatellite loci, recommended by the International Society of Animal Genetics (ISAG) [41], to characterise the genetic diversity and relationships between historical and modern populations of Russian local cattle breeds.

In microsatellite genotyping of DNA recovered from historical and ancient samples, amplification errors can occur, including allelic drop-out (ADO) and false alleles (FA) [37,38]. As the error rate may increase with a decrease in the amount and quality of DNA subjected to PCRs [77], we selected only the samples with dsDNA concentration higher than 1.04 ng/μL and OD260/OD280 ratio between 1.66 and 2.00 for the analysis. We generated consensus genotypes for 49 historical samples of cattle populations using a modified multiple-tube approach. Successfully amplified replicates were 97.67%, which corresponded to an amplification failure of 2.33%. We observed the previously reported [78,79] effect of target fragment size on PCR failure: the amplification of shorter fragments produced better results compared to that of longer ones (*r^2^* = 0.729, *p* = 0.01). Among the genotyping errors, ADO occurred more frequently compared to FA (2.41% and 0.75%, respectively), and agreed with the results of other studies of museum and ancient samples [79,80]. We observed a significant effect of allelic length on ADO rate (*r^2^* = 0.620, *p* = 0.05). For seven out of nine loci, the longer allele failed to amplify more often than the shorter allele; however, the differences were significant (*p* < 0.05) only in loci TGLA122 and TGLA126 (Appendix A).

Genetic diversity level, detected in historical populations, was comparable with those in modern populations of Kholmogor and Yaroslavl breeds (Table 1), and agreed with the results of a study of Italian medieval and modern Chianina and Romagnola cattle [20]. PCA plot (Figure 1), *F_ST_*-based network (Figure 2A), Jost’s D tree (Figure 2B), and STRUCTURE clustering (Figure 3) showed a clear differentiation of historical and modern Russian cattle from Holsteins, which confirmed the development of Russian black and white cattle breeds from different ancestral populations [13,14]. Neighboured positions of historical and modern populations of Kholmogor and Yaroslavl cattle on *F_ST_*-based and Jost’s D networks (Figure 2A,B), and the results of model-based clustering (Figure 3), confirm the maintenance of the historical genetic components in modern representatives of both breeds. Our findings are consistent with the results of previous studies of modern populations of these breeds performed using microsatellites [81] and genome-wide SNP markers [9,12,13], and confirm the need for conservation of Kholmogor and Yaroslavl cattle breeds as valuable national genetic resources.

The obtained result of the current study are clearly encouraging and are useful for managing modern populations of the studied breeds following the principle of sustainable animal breeding [82], while some limitations which are caused by the low amount and poor quality of DNA derived from historical samples, should be mentioned. Mostly because of the low genotyping quality of microsatellite loci with larger size of fragments, some of the available historical specimens were excluded from the analysis. As a result, three of the five studied historical populations were represented only by two or three samples. The small sample size might lead to some bias in the estimates of genetic diversity and breed relationships due to non-appearance among genotyped animals of some low-frequency alleles. To ratify our finding, additional studies of these and other available historical specimens using different molecular genetic approaches, including genome-wide SNP genotyping and whole-genome sequencing, are required.

## 5. Conclusions

By using a multiple-tube approach, we managed to extract consensus genotypes at nine microsatellite loci for historical specimens of the Russian local cattle breeds dated from the end of the 19th century to the first half of the 20th century. Conducted comparative studies detected the maintenance of the historical genetic components in modern Kholmogor and Yaroslavl cattle populations. Results of our research confirmed the great potential of studying historical DNA of livestock species for elucidation of breed history and selection of breeds and individuals for germplasm conservation.

## Figures and Tables

**Figure 1 genes-11-00940-f001:**
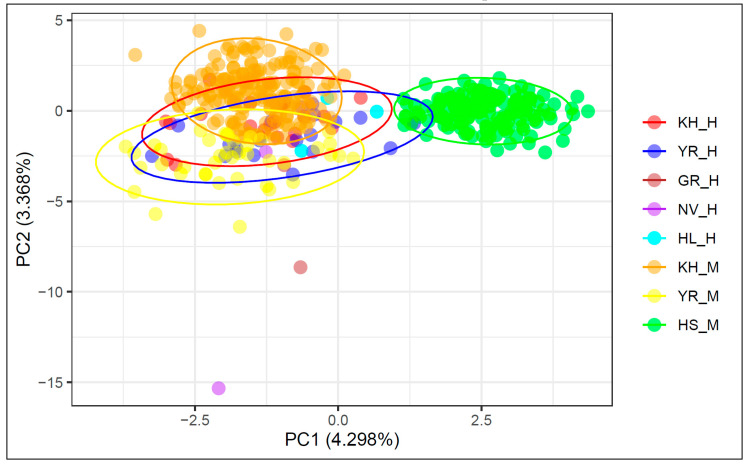
Principal component analysis (PCA) of historical and modern cattle populations based on nine STR markers. X-axis, principal component 1 (PC1); Y-axis, principal component 2 (PC2); KH_H, Kholmogor breed; YR_H, Yaroslavl breed; GR_H, Great Russian cattle; NV_H, Novgorod cattle; HL_H, Holland cattle; modern populations: KH_M, Kholmogor breed; YR_M, Yaroslavl breed; HS_M, Holsteins.

**Figure 2 genes-11-00940-f002:**
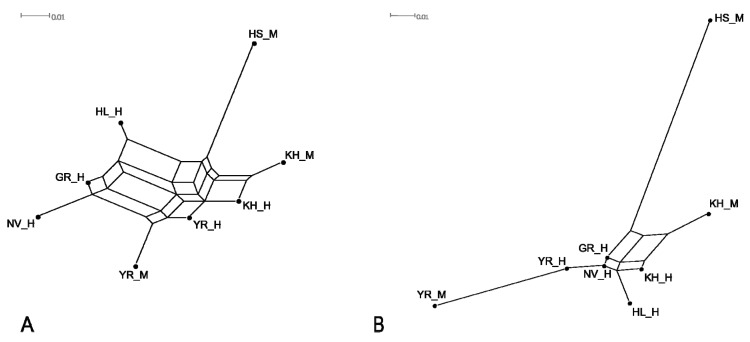
Neighbour-Net graphs based on *F_ST_* (**A**) and Jost’s D (**B**) genetic distances characterising genetic relationships between studied historical and modern cattle populations; KH_H, Kholmogor breed; YR_H, Yaroslavl breed; GR_H, Great Russian cattle; NV_H, Novgorod cattle; HL_H, Holland cattle; modern populations: KH_M, Kholmogor breed; YR_M, Yaroslavl breed; HS_M, Holsteins.

**Figure 3 genes-11-00940-f003:**
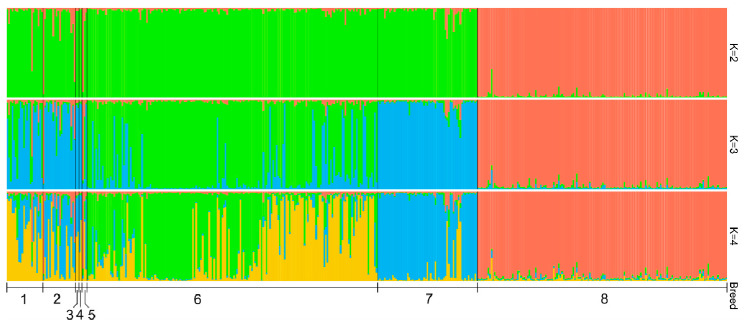
Genetic structure of historical and modern cattle populations revealed from the analysis of nine microsatellite loci. Historical populations: 1, Kholmogor breed; 2, Yaroslavl breed; 3, Great Russian cattle; 4, Novgorod cattle; and 5, Holland cattle; modern populations: 6, Kholmogor breed; 7, Yaroslavl breed; 8, Holsteins.

**Table 1 genes-11-00940-t001:** Observed allele ranges and distribution of genotyping errors among microsatellite loci in historical samples of studied cattle populations (%).

#	Locus	Observed Allele Ranges, bp ^a^	Number of Alleles	Amplification Failure, %	ADO Rate, %	FA Rate, %	ER, %
1	BM2113	121–141	10	0.00	1.29	0.37	1.47
2	BM1824	178–188	5	0.74	3.24	0.74	2.96
3	ETH10	211–225	8	2.94	3.23	0.38	2.65
4	ETH225	140–160	8	1.84	1.03	0.37	1.12
5	INRA023	198–216	9	5.88	2.29	1.17	3.13
6	SPS115	248–260	6	5.51	4.07	1.17	3.89
7	TGLA122	139–181	16	0.74	1.99	1.11	2.59
8	TGLA126	115–127	7	2.57	3.69	0.75	3.77
9	TGLA227	77–103	12	0.74	0.90	1.11	1.85
Mean value			9.00 ± 3.35	2.33 ± 0.31	2.35 ± 0.36	0.79 ± 0.18	2.59 ± 0.33

Locus, microsatellite locus; observed allele ranges, the limits of allelic lengths of studied microsatellite loci in historical samples analysed (^a^ allele sizes were standardised according to International Society of Animal Genetics (ISAG) International Bovine *(Bos Taurus)* short tandem repeat (STR) typing comparison test 2018–2019); amplification failure, the number of replicates with failure amplification from the total number of PCR replicates (%); ADO rate, the rate of allelic drop-out (%); FA rate, the rate of false alleles (%); and ER, overall error rate (%).

**Table 2 genes-11-00940-t002:** Summary statistics based on nine STR markers.

Population	*n*	*H_O_* (M ± SE)	*_U_H_E_* (M ± SE)	*A_R_* (M ± SE)	*_U_F_IS_* (CI)
KH_H	22	0.783 ± 0.030	0.726 ± 0.026	2.716 ± 0.092	−0.080 (−0.131; −0.029)
YR_H	20	0.789 ± 0.036	0.774 ± 0.023	2.893 ± 0.088	−0.020 (−0.099; 0.059)
GR_H	2	0.722 ± 0.121	0.852 ± 0.043	3.111 ± 0.261	0.167 (−0.118; 0.452)
NV_H	2	0.611 ± 0.111	0.722 ± 0.104	2.778 ± 0.324	0.137 (−0.088; 0.362)
HL_H	3	0.556 ± 0.111	0.644 ± 0.100	2.533 ± 0.268	0.147 (−0.023; 0.317)
KH_M	177	0.714 ± 0.027	0.708 ± 0.031	2.661 ± 0.106	−0.011 (−0.030; 0.008)
YR_M	61	0.674 ± 0.045	0.739 ± 0.032	2.758 ± 0.106	0.080 (−0.039; 0.199)
HS_M	152	0.705 ± 0.035	0.697 ± 0.031	2.616 ± 0.104	−0.011 (−0.045; 0.023)

*n*, number of individuals; *H_O_*, observed heterozygosity; *_U_H_E_*, unbiased expected heterozygosity; *A_R_*, rarefied allele richness [57]; *_U_F_IS_*, unbiased inbreeding coefficient; M, mean value; SE, standard error; CI 95%, range variation coefficient of *uFis* at a confidence interval of 95%; KH_H, Kholmogor breed; YR_H, Yaroslavl breed; GR_H, Great Russian cattle; NV_H, Novgorod cattle; HL_H, Holland cattle; modern populations: KH_M, Kholmogor breed; YR_M, Yaroslavl breed; HS_M, Holsteins.

**Table 3 genes-11-00940-t003:** Genetic distances between the studied populations based on the analysis of nine microsatellite loci.

Population	KH_H	YR_H	GR_H	NV_H	HL_H	KH_M	YR_M	HS_M
KH_H	-	0.017	0.072	0.068	0.060	0.018	0.047	0.076
YR_H	0.024	-	0.033	0.060	0.061	0.041	0.027	0.062
GR_H	0.025	0.003	-	−0.061	0.006	0.089	0.034	0.079
NV_H	0.008	0.034	−0.018	-	0.067	0.107	0.063	0.133
HL_H	0.021	0.031	−0.008	−0.001	-	0.064	0.082	0.085
KH_M	0.016	0.081	0.020	0.048	0.067	-	0.071	0.075
YR_M	0.070	0.056	0.001	0.045	0.123	0.174	-	0.105
HS_M	0.146	0.125	0.016	0.082	0.156	0.127	0.256	-

*F_ST_* values are presented above the diagonal; Jost’s D values are presented below the diagonal; KH_H, Kholmogor breed; YR_H, Yaroslavl breed; GR_H, Great Russian cattle; NV_H, Novgorod cattle; HL_H, Holland cattle; modern populations: KH_M, Kholmogor breed; YR_M, Yaroslavl breed; HS_M, Holsteins; table cells with the results of pairwise comparison within historical populations, as well as pairwise comparison within modern populations are shown by gray filing.

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
