# Peer review of "Genetic Diversity of Historical and Modern Populations of Russian Cattle Breeds Revealed by Microsatellite Analysis"

_genes, 2020, doi:10.3390/genes11080940_

Round 1
Reviewer 1 Report
The revised manuscript has been significantly improved. Additional work and analyses have been performed to account for false alleles and bias factors. However, critical limitation remains due to very parse genetic markers being used to infer genetic relationship between the historic and modern Russian cattle breeds. Sample size (2, 2 and 3) of 3 historic breeds is also a challenge. Otherwise, there are few minor corrections required as mentioned in below comments. As previously pointed out that the discussion of manuscript lacks any insights about the limitations (sample size and specimen availability, marker density) of this study and any other options to validate these results in future.
Line 5: Contribution of the newly added author are not provided in lines: 377-381
Line 37: Change “development” to “subsistence”.
Line 41: Change “worse adaptation ability” to “lack of adaptability”.
Line 44: Change “environment” to “conditions”, and elaborate what is “so on”?
Line 47: “sustainable breeding”?
Line 49: delete “breeds”.
Line 53: Change “;” to a full stop.
Line 54: Unpretentiousness?
Line 56: Change “several authors” to “numerous studies”.
Lines 56-57: Please provide numerical proportion of Holland cattle in both cases, high and low.
Line 59: homonymous?
Line 60-63: Too long sentence, unnecessary wordiness. Keep it simple with population numbers and years.
Lines 66-81: It is important to acknowledge previous research on various topics, however this section shows excessive use of references. Please use the most relevant, high quality and recent articles or preferable reviews to support the use of each type genetic markers.
Line 69: Change “single nucleotide polymorphism (SNP)” to “SNP”.
Line 71: “processes”?
Line 72: Change “control” to “testing”.
Line 73: Change “specimens, which were stored” to “specimens stored”
Line 81: Don’t understand this: “they enable the use of different molecular genetic tools for their analyses”?
Lines 96-98: Should briefly elaborate the advantages of multiple-tubes procedure of genotyping.
Line 184: Acronym “ERs” is unnecessary as elsewhere not used in text.
238: “NVGR_H”?
Lines 244-320: Try and be consistent to use either breed names or acronyms between the manuscript’s text and figures/tables.
322-358: Discussion is more like an introduction and partial repetition of results, however lacks any insights on its limitations and how future studies may be required to validate these findings in larger animals with high-density SNPs and genome sequencing.
Lines 354-358: Too long sentence and very convoluted.
Author Response
Dear reviewer,
We would like to thank you for careful reading of our manuscript and we value the comments received greatly, as they have been very useful to improve the paper. We agree with the comments and made appropriate corrections to the text. We marked all corrections in the text in red.
We would like to clarify some of the comments.
Line 47:“sustainable breeding”?
Reply: We used term sustainable breeding to emphasize the need to take care of animal genetic resources when breeding them with the aim of productive use. According to Gamborg and Sandøe (Livest. Prod. Sci., 2005, 92, pp. 221-231), sustainable animal breeding is “the extent to which animal breeding and reproduction, as managed by professional organizations, contribute to the maintenance and good care of animal genetic resources for future generations”.
Line 54: Unpretentiousness?
Reply: We changed it to adaptability.
Lines 56-57: Please provide numerical proportion of Holland cattle in both cases, high and low.
Reply: The first import of Dutch cattle to Kholmogor district dates back to 1725, but there is no information about the number of imported livestock and its impact on the local livestock. It is known, that in 1765 was the cattle importation from Holstein, in 1805 were imported two cows from Amsterdam, in 1818 – 4 bulls and 12 cows from Holland. Then in 1865, the Ministry of state property distributed 19 bulls and five cows of Holland breed between the peasants. However, there is no precise information concerning the contribution of Holland cattle to the development of the population of Kholmogor cattle. According to some sources, the peasants did not actively use Holland bulls in breeding, as the resulting crosses with Kholmogor cattle were inferior in their characteristics to purebred Kholmogor cattle. Other sources indicate the existence of three types of cattle in Kholmogor district at the beginning of the 20th century. The first one is the Holland type, which originates from imported cattle. A small number of animals of this type was found in some peasant farms. The second type is the local Kholmogor cattle, which are larger comparing to Holland cattle. The third type is the relatively small animals known as “simple Russian cattle”. Most of the population of Kholmogor cattle in that time was represented by various crosses with different contributions of the three types mentioned above. However, the precise distribution between different types and crosses is not documented.
Line 60-63: Too long sentence, unnecessary wordiness. Keep it simple with population numbers and years.
Reply: We shortened the sentence according to the suggestion
Lines 66-81: It is important to acknowledge previous research on various topics, however this section shows excessive use of references. Please use the most relevant, high quality and recent articles or preferable reviews to support the use of each type genetic markers.
Reply: We kept only the most relevant references and replaced other one by review.
Line 81: Don’t understand this: “they enable the use of different molecular genetic tools for their analyses”?
Reply: We rephrased the sentence to “The recent success in development of protocols for effective DNA extraction from ancient and historical samples enable their molecular genetic analysis using different tools”
Lines 96-98: Should briefly elaborate the advantages of multiple-tubes procedure of genotyping.
Reply: We included in the text the following sentence: “Comparing to the standard one-tube procedure, the multiple-tube approach has the advantage of providing a quantitative measure of the degree of support for each possible genotype”.
322-358: Discussion is more like an introduction and partial repetition of results, however lacks any insights on its limitations and how future studies may be required to validate these findings in larger animals with high-density SNPs and genome sequencing.
Reply: We expanded the discussion section according to the reviewer’s suggestion.
Reviewer 2 Report
I read the MS accurately by Abdelmanova and co-authors. It describes the genetic diversity of historical and modern populations of Russian cattle breeds analyzed using a microsatellite approach. I suppose that the MS was already revised previously because it seems mature for publication. However, there are some modifications that I can suggest that may improve the readability of the MS.
- Abstract: Re-write abstract focusing on results and not on methods. I think that this paper did not “..reconstruct the history of local cattle husbandry” but, how defined by the title, it describes the “Genetic diversity of historical and modern populations”.
- Paragraph 2.2 - method about ancient DNA should be more accurate about the procedure to avoid cross contaminations, blank extract, and if they performed DNA extraction in another laboratory.
- Paragraph 2.3 - In order to define the quality of microsatellite loci, selected authors should add information about the number of alleles previously described in the literature and not only found in samples.
- Paragraph 2.4 - describe better all index chosen for statistical analysis, their significance and range.
- Paragraph 3.2 - describe the variability of populations, not give only table, in table 2 legend re-write the complete name of all populations.
- Figure 1 - use circle with opacity <50% in order to understand the density of overlapped samples. In legend, write the names of different populations again. Use appropriate font and dimension for figure legend.
- Line 275-6 delete this sentence and modify sentence line 281-2 “Genetic relationships identified between the studied breeds based on both pairwise Jost’s D and FST values in most cases were similar (Table 3).
- Figure 2 - letters “A” and “B” are too big.
- The discussion should be enlarged considering the importance of historical population to maintain biodiversity; the comparison of results not only with mtDNA but also on previous paper on SNPs cited in introduction and future remarks. Maybe authors can define some management suggestions too.
Author Response
Dear reviewer,
We would like to thank you for taking time in consideration of our manuscript and we value the comments received greatly, as they have been very useful to improve the paper.
We agree with the comments and made appropriate corrections to the text. We marked all corrections in the text in red.
We would like to explain some of the comments.
Abstract: Re-write abstract focusing on results and not on methods. I think that this paper did not “..reconstruct the history of local cattle husbandry” but, how defined by the title, it describes the “Genetic diversity of historical and modern populations”.
Reply: We rewrote abstract considering the reviewer’s suggestions.
Paragraph 2.2 - method about ancient DNA should be more accurate about the procedure to avoid cross contaminations, blank extract, and if they performed DNA extraction in another laboratory.
Reply: we included the information about both the negative control (“reagent blank”) in the text of paragraph 2.2 of our manuscript, and the negative control of PCR in the text of paragraph 2.3.
Paragraph 2.3 - In order to define the quality of microsatellite loci, selected authors should add information about the number of alleles previously described in the literature and not only found in samples.
Reply: we added the information concerning the number of alleles identified at each microsatellite loci to the text of this paragraph.
Paragraph 2.4 - describe better all index chosen for statistical analysis, their significance and range.
Reply: calculations of quality indices (QIs) are described in details in paragraph 2.3 because it related to the estimation of consensus genotypes; the threshold for QIs was set at 0.75 as it is mentioned in paragraph 2.4. We added the data about observed ranges of QIs in paragraph 3.1 and added the data on QI values at each microsatellite locus in Supplementary materials, Tab. S2. We calculated the probability values of correct genotyping and included this data in Tab. S2.
Paragraph 3.2 - describe the variability of populations, not give only table, in table 2 legend re-write the complete name of all populations.
Reply: we expanded the description of the variability parameters; we added in table 2 the legend with the complete names of all populations.
Figure 1 - use circle with opacity <50% in order to understand the density of overlapped samples. In legend, write the names of different populations again. Use appropriate font and dimension for figure legend.
Reply: we replaced the filling of circles from nontransparent to half-transparent, added the names of different populations to legend and replaced the font.
The discussion should be enlarged considering the importance of historical population to maintain biodiversity; the comparison of results not only with mtDNA but also on previous paper on SNPs cited in introduction and future remarks. Maybe authors can define some management suggestions too.
Reply: we expanded the discussion section according to the suggestions of the reviewer.
This manuscript is a resubmission of an earlier submission. The following is a list of the peer review reports and author responses from that submission.
Round 1
Reviewer 1 Report
Comments to authors:
In their work, Abdelmanova et al. carry out a molecular investigation of Russian native cattle based on microsatellite DNA loci by comparing historical and modern representatives of different breeds. The purpose is to quantify the amount of genetic diversity held by such breeds through time and explore the legacy among them. Even if not novel in its methods, the study is certainly valuable since it represents a preliminary molecular characterization of valuable agrobiodiversity resources based on widely used markers, thus producing useful data for future comparative studies. Even if these days SNPs studies using tens of hundred thousand loci are increasingly popular, microsatellites keep maintaining their importance, for instance, in kinship studies and pedigree reconstructions.
That said, I was puzzled by the decision of the authors of not using (or not mentioning and comparing previous works if pertinent data are already available – maybe the only exception being citation n. 6) mitochondrial DNA (mtDNA), which represent the marker of choice to study the “legacy” of different populations (in this case breeds) through time and space, just like in phylogeographic studies. Also, mtDNA is present in much higher copy number and, for this reason, straightforward to extract and process then microsatellite in case of low-quality DNA samples. This brings me to face with the most critical passage of the entire work: the total absence of PCR replicates in case of DNA amplification from historical specimens. When working with museum specimens and historical DNA as a whole, replicates to confirm genotypes by building consensus accounting for false alleles and allelic dropout are a must, otherwise any result is not reliable. In this MS, I have not found any trace of these replicates being carried out, nor I can see a table with consensus genotypes and allelic ranges (just the number of alleles). Microsatellites are definitively much more variable than mtDNA and, as such, are more informative of the amount of genetic diversity that has been preserved through time or gone lost. On the other hand, mtDNA is a complimentary and fundamental source of information.
That said, the authors should:
- carry out PCR replicates for historical specimens and extract consensus genotypes;
- redo the analyses of modern vs historical breeds based on these consensus genotypes;
- provide the table with consensus genotypes to the readership (it can be either annexed as supplementary information or deposited in open access repositories such as Dryad);
- produce at least a subset of mtDNA data (can be the Cytochrome-b) including representatives of all historical and modern breeds included here and a mtDNA network which is way more indicated in cases where the relationships between pairs of taxa may be other than dichotomical (as they are instead forced to be in a phylogenetic tree).
On top of this, the paper should be presented in a more concise, direct and organized fashion. There are some relevant conceptual biases, some information is not relevant, while fundamental pieces of information are missing, like the bibliographic references of the microsatellites loci being used, what the numbers introduced with ± in Table 1 stand for, and allelic ranges. Also, it is not used how many K clusters were considered: in the methods I understand that only K = 2 was explored, but in the Results it seems that multiple K were explored instead. Moreover – and this is not a marginal issue – the MS is full of convoluted sentences and wordiness is an issue. The use of the articles and preposition (for instance “between” vs “among”, “for” vs “with”) is not consistent; many indices (e.g., FST) are not written properly. Repetitions or inessential details should be avoided. Large parts of the discussion are a mere repetition of the results. Comments, corrections and suggestions are conveniently reported in the MS as point-by-point comments. I warmly recommend getting a native speaker going through the MS before a new submission is considered.
Reviewer 2 Report
This is a very interesting effort to investigate historical and modern samples of Russian breeds. In these times of convenient and low cost genome-wide sequencing and genotyping, choosing a few microsatellites (n=9) gives very low power to the analyses in this study. Although the authors have highlighted the limitations of using historical samples, recent investigations have been able to generate high-density genomic data. I suggest the authors to consult the following article and appropriately update the manuscript.
McHugo et al. 2019 (BMC Biology). Unlocking the origins and biology of domestic animals using ancient DNA and paleogenomics. https://bmcbiol.biomedcentral.com/articles/10.1186/s12915-019-0724-7
In addition, following comments may help improve some sections of the manuscript.
Lines 25-27: In abstract, please report the findings and insights not the statements about analyses.
Line 37: “adaptation”, I think otherwise because local adaptation is the key point in favour of indigenous livestock against the exogenous as you rightly mentioned in lines 39-40.
Line 44: change “animal breeds” to “livestock breeds”
Each of the participating breeds should be briefly introduced for their unique phenotypic and production features as well as some historic outlook about their crossbreeding and introgression in the Introduction section (see a comment to move a paragraph out of Discussion).
Lines 80-86: Breed abbreviations are too long. I suggest to use two letter abbreviations, eg., KM instead of KHLM. Moreover, keep the sequence while reporting historic and modern samples, ie., Kholmogor before Yaroslavl.
Line 122: Provide reference for ISAG.
Lines 123-136: These lines belong to Results. Otherwise, I suggest removing such information as these are preliminary assessments and only report the markers which were successful.
Line 158: what is “rarified” allele richness?
Lines 160-162, 174-176, 209-211, 218-220, 352-354: Don’t need to repeat defining the breed abbreviations.
Line 182: May replace “group size” with “sample size (n=2)”.
Lines 255-258: Use either breed names or abbreviations to denote each number in the graph – Figure 3.
Lines 260-272: This paragraph should be moved to introduction because this is what authors already know and it should be used as base to evaluate these “facts” through this study.
Lines 291-303: These are results, a bit repetition of what has been stated already.
Line 318: “dishonest cattle sellers”. Seems a little harsh statement, may be true. Lack of awareness both by sellers and buyers as well as availability of required numbers of pure-bred may be were some other issues.
The discussion section lacks any insights about the limitations (sample size and availability, marker density) of this study. Also, there are no concluding recommendations about which breeds should be conserved.